# Strain Engineering of Intrinsic Ferromagnetism in 2D van der Waals Materials

**DOI:** 10.3390/nano13162378

**Published:** 2023-08-19

**Authors:** Hongtao Ren, Gang Xiang

**Affiliations:** 1School of Materials Science and Engineering, Liaocheng University, Hunan Road No. 1, Liaocheng 252000, China; 2College of Physics, Sichuan University, Wangjiang Road No. 29, Chengdu 610064, China

**Keywords:** strain engineering, ferromagnetism, transition metal trihalides, transition metal chalcogenides, transition metal phosphorous chalcogenides, wrinkle, flexible substrates, lattice mismatch, spintronics, straintronics

## Abstract

Since the discovery of the low-temperature, long-range ferromagnetic order in monolayers Cr_2_Ge_2_Te_6_ and CrI_3_, many efforts have been made to achieve a room temperature (RT) ferromagnet. The outstanding deformation ability of two-dimensional (2D) materials provides an exciting way to mediate their intrinsic ferromagnetism (FM) with strain engineering. Here, we summarize the recent progress of strain engineering of intrinsic FM in 2D van der Waals materials. First, we introduce how to explain the strain-mediated intrinsic FM on Cr-based and Fe-based 2D van der Waals materials through ab initio Density functional theory (DFT), and how to calculate magnetic anisotropy energy (MAE) and Curie temperature (*T_C_*) from the interlayer exchange coupling J. Subsequently, we focus on numerous attempts to apply strain to 2D materials in experiments, including wrinkle-induced strain, flexible substrate bending or stretching, lattice mismatch, electrostatic force and field-cooling. Last, we emphasize that this field is still in early stages, and there are many challenges that need to be overcome. More importantly, strengthening the guideline of strain-mediated FM in 2D van der Waals materials will promote the development of spintronics and straintronics.

## 1. Introduction

According to the *Mermin*–*Wanger*–*Hohenberg* theory [1,2], thermal fluctuations can destroy the long-range magnetic order of 2D systems at finite temperatures. However, the anisotropy of the system suppresses thermal disturbances by opening the gap in the spin-wave spectrum [3,4,5,6]. Furthermore, spin orbit coupling (SOC) can stabilize the long-range magnetic order in 2D systems by contributing to magnetic anisotropy. After the discovery of the low-temperature, long-range ferromagnetic order in monolayers Cr_2_Ge_2_Te_6_ and CrI_3_ [7,8], many efforts have been made to achieve a room temperature (RT) ferromagnet. Indeed, strain engineering [9,10,11,12,13,14,15,16,17,18,19,20,21,22,23,24,25] is a very important strategy for mediating material properties, including optoelectronic [9,10,13,14,15,21], electrocatalytic [11,16,22,23,24], and magnetic properties [15,19,21,24,25,26,27,28]. Since Novoselov et al. [29] obtained a stable monolayer graphene in the laboratory in 2004, further research gradually revealed that 2D materials, such as MoS_2_, could withstand up to 20% strain [30,31,32,33]. However, it was very difficult to directly apply strain to 2D materials in experiments, which made strain-controlled performance largely remain in theoretical study. This was because by changing lattice parameters, strain could be easily applied to the lattice of 2D materials. Specifically, the study on strain-mediated magnetism in 2D materials, especially in 2D van der Waals materials with intrinsic long-range FM order, was focused on theoretical calculation. More interestingly, the Bi_2_Te_3_|Fe_3_GeTe_2_ heterostructure related to strain [34,35] was designed to increase the Curie temperature (*T_C_*) due to the proximity effect [36,37,38,39,40].

Very recently, some significant progress was also made in the field of experimental research [27,28,41,42,43,44,45,46,47,48]. In Figure 1, we summarize the recent progress of strain engineering of intrinsic ferromagnetism (FM) in 2D van der Waals materials. First, we introduce how to explain the strain-mediated intrinsic FM on Cr-based and Fe-based 2D van der Waals materials with long-range FM order through ab initio Density functional theory (DFT), and how to calculate magnetic anisotropy energy (MAE) and Curie temperature from the interlayer exchange coupling J. Subsequently, we focus on numerous attempts to apply strain to 2D materials in experiments, including wrinkle-induced strain, flexible substrate bending or stretching, lattice mismatch, electrostatic force and field-cooling. Last, we emphasize that this field is still in early stages, and there are many challenges that must be overcome. More importantly, strengthening the guideline of strain-mediated FM in 2D van der Waals materials will promote the development of spintronics [6,49,50,51,52,53,54,55,56,57] and straintronics [12,19]. As shown in Figure 1, we summarize three different kinds of 2D materials with intrinsic long-range FM order, including transition metal trihalides (CrCl_3_, CrBr_3_ and CrI_3_) [5,58,59,60,61,62,63,64,65,66,67,68,69,70,71,72], transition metals chalcogenides (Cr_2_Ge_2_Te_6_, Fe_n_GeTe_2_ and CrTe_2_) [42,43,44,45,46,73,74,75,76,77,78,79,80,81,82,83,84,85,86,87], and transition metal phosphorous chalcogenides (AgVP_2_Se_6_, CrPS_4_) [88,89,90,91].

## 2. Theoretical Calculations

In order to understand the essence of 2D ferromagnetism, ab initio Density functional theory, including linear density approach (LDA) [59], local spin density approximation (LSDA) [92], the generalized gradient approach (GGA) [61,93,94], and DFT + U [95,96], was often used to calculate the electronic structure of the system as a starting point. Moreover, the interlayer exchange coupling J was closely related to magnetic anisotropy, and it would also be used to calculate *T_C_* [86,97,98,99,100]. The mean field theory [97,101] would roughly estimate *T_C_* but, often, *T_C_* overestimated it. Although the random phase approximation (RPA) could more accurately estimate *T_C_* of three-dimensional (3D) materials, it may fail in 2D systems with large anisotropy. Notably, classic Monte Carlo (MC) [6,97] simulations can also describe the critical temperature.

### 2.1. Cr-Based 2D van der Waals Materials

#### 2.1.1. CrCl_3_

Unlike bulk materials, 2D materials can sustain larger strains [33,102]. Similarly, single-layer transition metal trihalides (MX_3_, I, Cl and Br) can also withstand a strain of about 10% [58]. As a typical example, Yan et al. [59] studied the biaxial strain dependence magnetic anisotropy energy (MAE) of the 2D monolayer CrCl_3_ (Figure 2a–d). When the compressive strain reached 2.5%, a phase transition from antiferromagnetism (AFM) to FM occurred (Figure 2e). In addition, when tensile strain was 2.4%, the maximum Curie temperature (*T_C_*) reached 39 K. The MAE in the unstrained monolayer was positive, indicating the spins of Cr atoms were off-plane (Figure 2f). 

After applying the strain to the lattice, Cl atoms adjusted their position to minimize the lattice distortion energy at this strain (Figure 2g). After generating a specific structure, the energy difference between the ferromagnetic and antiferromagnetic states was calculated. Mapping this energy difference to the Hamiltonian (1), Dupont et al. [63] obtained the nearest-neighbor exchange coupling J and magnetic anisotropy *K*, as shown in Figure 2h,i.
(1)$$\hat{H}=J_{\varepsilon }\sum_{\langle \vec{r},{\vec{r}}^{\prime }\rangle }{\hat{\vec{S}}}_{\vec{r}}\cdot {\hat{\vec{S}}}_{{\vec{r}}^{\prime }}+K_{\varepsilon }\sum_{\vec{r}}{\left(\hat{\vec{S}}{}_{\vec{r}}^{z}\right)}^{2}\;$$

Note that: the spin value of S was 3/2 in the above equation. 

As the strain evolved from compression to tension, the system sequentially exhibited the BKT (Berezinskii–Kosterlitz–Thouless) quasi-long-range order (LRO) phase, AFM Ising and FM Ising by QMC (Quantum Monte Carlo) simulations. Although theoretical and experimental studies have been conducted on monolayer and bulk CrCl_3_ materials, research on their multilayer structures, including bilayers and trilayers systems, was very limited. Ebrahimian et al. [64] found that biaxial strains could also achieve a phase transition from AFM to FM (Figure 2k–n). In addition, the magnetic anisotropy could be mediated by the strain.

#### 2.1.2. CrBr_3_

More interestingly, the Curie temperature of monolayer CrBr_3_ could be increased to 314 K by doping [103], which was between CrCl_3_ (323 K) and CrI_3_ (293 K). Although both hole doping and electron doping could enhance ferromagnetic coupling, the effect of hole doping was better at the same doping concentration. After applying biaxial strain to a unit cell, its magnetic moment remained unchanged, which indicated that the biaxial strain could not effectively enhance the ferromagnetic coupling of monolayer CrX_3_. In addition, Webster et al. [59] found that applying a tensile strain of 2.1% could increase the *T_C_* to 44 K, which was about 5 K higher than when no strain was applied. However, at a compressive strain of −4.1%, a FM to AFM phase transition, similar to CrCl_3_ [59,63,64] and CrI_3_ [59,70,71,72], also occurred.

#### 2.1.3. CrI_3_

Unlike monolayer CrCl_3_, the electronic bandgap of monolayer CrI_3_ remained almost unchanged after applying biaxial tensile strain; after applying biaxial compressive strain, the electronic bandgap decreased significantly and MAE increased significantly. When compressive strain reached 5%, MAE increased by 47% [59]. Continuing to increase the strain (−5.7%) resulted in a phase transition from FM to AFM. Similar to CrCl_3_, Wu et al. [69] also found that the CrI_3_ monolayer underwent a complex phase transition from magnetic metals, half-metal, half-semiconductor to magnetic semiconductor as the strain evolved from compression (−15%) to tension (10%). 

As a typical example, Vishkayi et al. [71] investigated the effects of biaxial and uniaxial strain on the magnetism of monolayer CrI_3_ (Figure 3a–e). A similar phase transition from FM to AFM [59] was also observed when a compressive strain greater than 7% was applied (Figure 3d). As the strain increased, the electronic bandgap showed an opposite trend when applying compressive or tensile strain (Figure 3d). Interestingly, uniaxial strain enhanced the nearest neighbor, Dzialoshinskii–Moriya (DM) interaction, by breaking the inversion symmetry, and its effect was stronger than biaxial strain.

The phase transition from FM to AFM under compressive strain (−3% or −5%) had also been discovered in the CrI_3_ bilayer [70], similar to other previously reported systems [59,71]. In addition, Safi et al. [72] found that the phase transition occurred at −6% compressive strain. More importantly, they also discovered a second phase transition point from FM to AFM, which occurred near −2.5% compressive strain.

#### 2.1.4. CrTe_2_

Guo et al. [104]. found that applying uniaxial or biaxial tensile strain to monolayer CrTe_2_ in the *T* phase did not cause a phase transition from direct to indirect bandgap. Under biaxial strain, the CrTe_2_ monolayer [105] underwent phase transformation at −1% compressive strain in Figure 4a–k. 

Interestingly, the CrTe_2_ monolayer with strain-free was a FM state. After the tensile strain was applied, its *T_C_* could rise to 1022.8 K [106], and the magnetic moment of Cr atom increased linearly, which may be caused by the increase of the density of states at Fermi energy *N_EF_*. Magnetic anisotropy exhibited a different sensitivity to uniaxial and biaxial strain, as shown in Figure 4l. A monolayer was more sensitive to tensile strain, while a bilayer was more sensitive to compressive strain, and bulk was insensitive to the applied strain. Furthermore, band filling [108] of the monolayer also underwent a transition from out-of-plane to in-plane, while the bilayer and bulk did not undergo this transition (Figure 4m). More interestingly, the strain had the greatest impact on the dihedral angle $\theta_{D}$, followed by the effect of Cr-Te-Cr bond length (Figure 4n). Notably, the charge density wave (CDW) phase [107] promoted greater stability of the long-range FM order.

#### 2.1.5. Cr_2_Ge_2_Te_6_

As early as 2014, Li et al. [73] predicted that Cr_2_X_2_Te_6_ (X = Si, Ge), a layered crystal with intrinsic FM, could be obtained experimentally by exfoliation. Furthermore, the *T_C_* of Cr_2_Ge_2_Te_6_ (Cr_2_Si_2_Te_6_) was increased by 51.7(56) K under an isotropic tensile strain. Similarly, Dong et al. [75] also found that Cr_2_Ge_2_Se_6_ and Cr_2_Ge_2_Te_6_ were a stable FM semiconductor, as shown in Figure 5a–e. 

Through Monte Carlo simulation, it was found that the *T_C_* of Cr_2_Ge_2_Se_6_ (144 K) was five times that of Cr_2_Ge_2_Te_6_ (30 K) (Figure 5d). Interestingly, when compressive strain was applied, the *T_C_* decreased, and even at 2% strain, the phase became AFM. After applying tensile strain, the *T_C_* would be increased, even up to 500 K. Notably, the *T_C_* of Cr_2_Ge_2_Se_6_ with 5% strain was always higher than RT (Figure 5e). To our knowledge, 2D Cr_2_Ge_2_Se_6_ has not yet been prepared experimentally.

As a member of Cr-based materials, 2D CrPbTe_3_ (CPT) has a stable monolayer structure and a higher *T_C_*, as shown in Figure 5f. Similar to Cr_2_Ge_2_Se_6_, its *T_C_* (110 K) gradually increased from compressive strain (61 K) to tensile strain (150 K) (Figure 5g–j). In addition, strain can also induce spin reorientation from the in-plane to the out-of-plane.

#### 2.1.6. CrPS_4_

AgVP_2_Se_6_, as a typical quaternary FM semiconductor, was synthesized by heating the elements in evacuated silica tubes as early as 1988 [88]. Ouvrard et al. found that its polycrystalline powder showed low-temperature FM, high-temperature PM and *T_C_* of about 29 K. Recently, single crystal AgVP_2_Se_6_ samples [90] exhibited better stability than MX_3_ materials. As another typical example, single crystal CrPS_4_ [91] was obtained by the chemical vapor transport method and mechanical exfoliation. Interestingly, odd layers were ferromagnetic at lower temperatures, while even layers were antiferromagnetic, as shown in Figure 6. To the best our knowledge, there were currently no reports on regulating the magnetic properties of AgVP_2_Se_6_ and CrPS_4_ through strain engineering.

### 2.2. Fe-Based 2D van der Waals Materials

The bulk Fe_3_GeTe_2_ (FGT) with the hexagonal platelets was first synthesized by a direct solid-state reaction (SSR) [81]. Zhuang et al. [82] found that applying biaxial strain on single-layer metallic Fe_3_GeTe_2_ could enhance the MAE and total magnetic moment (Figure 7a–d). When 2% tensile strain was applied, its MAE increased by 50%; After the applied strain changed from compression to tension, the total magnetic moment also increased with the increase of strain. Through DFT calculations, Hu et al. [85] discovered that monolayer FGT was a metallic type ferromagnet, with its magnetic moment mainly localized on Fe atoms and its metallicity mainly derived from the Fe *d* orbitals in Figure 7e–g. 

After applying the biaxial strain shown in Figure 7h, there was a significant change in the spin-density distributions of the FGT monolayer. The magnetism of this system mainly came from Fe atoms (Figure 7i), especially Fe3 atoms at the inequivalent site (Figure 7f), with little contribution from Ge and Te atoms (Figure 7j). Due to the Poisson effect, when stretched in the in-plane direction, it contracted in the out of plane direction. Therefore, as the bond lengths of Fe1-Te and Fe3-Te were reduced in the plane, the bond lengths of Fe3-Te in the vertical direction were increased (Figure 7k). Biaxial strain caused a change in bond length, thereby promoting charge transfer within the monolayer (Figure 7l). When the biaxial strain increased from −5% to 5%, the spin splitting of the Fe1 3d orbital near the Fermi level became larger, as shown in Figure 7m; after the strain was applied, the spin polarization of Fe3 atoms would increase in Figure 7n, which would cause the magnetic moment to increase. In conclusion, biaxial strain-mediated FM in the FGT monolayer was closely related to the charge transfer between Fe1 and Te atoms.

Previous studies have mainly focused on biaxial strain-mediated FM in the FGT monolayers [82,85], without studying the modulation mechanism of uniaxial strain on FM. As a typical example, Zhu et el. [86] investigated the uniaxial (*ε_a_* and *ε_c_*) and isotropic (*ε_iso._*) strain modulation of the exchange coupling constant J*_ij_* and *T_C_* in FGT. As shown in Figure 8, three different magnetic configurations (FM, AFM1, and AFM2) were considered. The total energy of AFM2 was much higher than those of FM and AFM1. 

The system exhibited a phase transition from AFM to FM, similar to the modulation result of biaxial strain [59]. Interestingly, when the strain (*ε_a_*) along the a-axis was applied, the phase transition occurred at a tensile of ~4% or compressive of ~2%; when the strain (*ε_c_*) along the c-axis strain was applied, it occurred at a compressive of ~8–10%. Furthermore, the strain-mediated variation of J*_ij_^total^* is shown in Figure 8g–i. *T_C_* could be estimated by the following equation [86,98,99]:(2)$$T_{C}=\frac{2}{3K_{B}}J_{max}\;$$

Note that $J_{max}$ is the maximal eigenvalue of the matrix consisting of the exchange coupling between different atoms [86,109]. After applying the uniaxial strain, the $T_{C}$ could be raised to room temperature (Figure 8k–m). When isotropic strain is applied, the $T_{C}$ exhibited complex nonlinear changes and failed to reach room temperature (Figure 8m). Their results demonstrated that applying uniaxial strain was an effective way to elevate the $T_{C}$.

Regarding another interesting compound, Fe_5_GeTe_2_ [110,111], recent experiments have found that it has a higher T_C_. However, little is known about the electronic and magnetic properties of its monolayer. Joe et al. [83] found that the magnetism of bulk and monolayer metallic Fe_5_GeTe_2_ originated from Fe d orbital. Moreover, biaxial strain could also enhance the Fe magnetic moment from 1.65 µ_B_ to 2.66 µ_B_.

## 3. Introducing Strain in 2D van der Waals Materials

### 3.1. Wrinkle-Induced Strain

The previously discovered strain-mediated FM phenomena were all in 2D materials without intrinsic long-range magnetic order [27,28,41]. Very recently, Seidel et al. [42] found a clear dependence of *T_C_* on the strain state and the thickness of Cr_2_Ge_2_T_6_ (CGT). The layered CGT lattice with intrinsic magnetism was shown in Figure 9a. A stripe domain structure was observed with MFM (Figure 9b). After buckling, the wrinkled area had a higher *T_C_* than the flat area on (Figure 9c). DFT calculations indicated that the strain could elevate the *T_C_* in monolayer and bilayer Cr_2_Ge_2_T_6_ (Figure 9d–g). In addition, the strain of three different wrinkles in Figure 9h was confirmed by COMSOL simulations. To the best of our knowledge, this was the first time that the strain distribution in CGT wrinkles was mapped with the COMSOL simulation. Furthermore, the magnetic signals could be observed at the wrinkled regions at RT through MFM. 

### 3.2. Bending or Pre-Stretching Flexible Substrates

Similar to pre-stretched flexible substrates, including Gel-Film [30], polydimethylsil- oxane (PDMS) [30,112,113,114,115,116,117], polyethyleneterephthalate (PET) [118], polyimide (PI) [47,119,120], and polyvinyl alcohol (PVA) [121,122], strain can also be introduced into 2D materials by bending flexible substrates. As a typical example, Yan et al. [121] designed a novel polymer-buried strategy to apply tensile strain on Fe_3_GeTe_2_ (FGT) nanoflakes (Figure 10a). Firstly, PVA was spin-coated onto pre-stretched mechanical exfoliated FTG nanosheets. Then, the PET sheet was attached to the surface of the PVA film and peeled off with tweezers. The complex film (FGT/PVA/PET) was placed into a non-magnetic plastic tube and then the tensile strain was applied on the FGT nanosheets through a three-point fixing device. As the applied strain gradually increased, the sample transitioned from its original PM state to FM states, as shown in Figure 10b,c. When the strain reached 4.7%, the hysteresis loop evolved from soft magnet to hard magnet; however, when the strain reached 7.0%, the hysteresis loop displayed an opposite evolution trend. Moreover, the *M_s_* and *H_c_* were very close to those found for the case of tensile strain. Unlike zero-strain samples, the sample with a 3.4% tensile strain always exhibited centrosymmetry and no exchange bias (Figure 10d,e). The hidden AFM state and interface–exchange interaction could be revealed by controlling the strain. The frequency shift was inverted after relaxing the strain, indicating the transition from FM state to PM state (Figure 10f,g). As the strain increased, the *T_C_* was elevated above RT in Figure 10h,i. In short, strain engineering is an efficient way to increase *T_C_* compared to other methods (Figure 10j).

As another typical example, Miao et al. [47] found that when 0.32% uniaxial tensile strain was applied to the FGT nano sheet, its *H_c_* increased by more than 150%. In order to apply strain, they directly exfoliated the FGT nanosheets onto the PI film. By controlling the distance of the needle tip pushed at the center of the substrate, the PI film was bent to apply different uniaxial tensile strains to the FGT sample. The difference of magnetic anisotropy energy was attributed to the strain-mediated FM. More importantly, they realized a magnetization reversal with the limited strain. Similarly, Xu et al. [48] also observed a reversible phase transition from AFM to FM in a CrSBr nanoflake at cryogenic temperature. Their strain equipment consisted of three piezoelectric actuators glued to a titanium flexure element. By cleaving a silicon substrate to form a micrometer-scale gap, the sample was suspended at the gaps. Furthermore, a piezo voltage was used to continuously apply strain reversibly to the CrSBr flake.

### 3.3. Lattice Mismatch

High-quality, single-crystalline Fe_4_GeTe_2_ thin films [43] were grown on sapphire substrate by molecular beam epitaxy (MBE). As shown in Figure 11a, the thin films exhibited a rhombohedral structure. During the MBE epitaxy preparation, due to a lattice mismatch of about 20% between the Fe_4_GeTe_2_ sample and the sapphire substrate, the lattice rotated 30° to form a perfect single crystal. 

However, there was a tensile strain of about 2% inside the Fe_4_GeTe_2_ film. The in-plane *M*-*T* curve of the 16 nm FGT film in Figure 11b indicated that its *T_C_* was close to RT. The results of XRD (Figure 11c) and HRTEM (Figure 11d) indicated that the FGT film had a perfect single crystal, and the atomic ratio of the Fe:Ge:Te element was about 4:1:2. At 300 K, a 4 nm film exhibited robust in-plane magnetic anisotropy, as shown in Figure 11e. By fitting the *M_r_*-*T* curve in Figure 11f, the *T_C_* could be deduced as high as 530 K. Furthermore, in Figure 11, it was confirmed through *M*-*H* loops that the few layered FGT sample exhibited high-temperature FM. In addition, the thickness dependence of *T_C_* showed that when the thickness was thinner, its *T_C_* was higher, as shown in Figure 11h,i. Notably, MBE-prepared FGT had the higher *T_C_* and maintains a relatively high-magnetic anisotropy (Figure 11j). Through DFT calculations, it was found that a 2% tensile strain was not the reason for the enhanced *T_C_*.

### 3.4. Electrostatic Force

Considering the abnormally large mechanical response of Cr_2_Ge_2_Te_6_ (CGT) thin films caused by strong magnetostriction at *T_C_*, it was expected that this reverse effect could achieve strain-controlled *T_C_*. As shown in Figure 12a–c, the electrostatic strain-tuning *T_C_* was realized in a suspended Cr_2_Ge_2_Te_6_/WSe_2_ heterostructure [44]. After applying a gate voltage *V_g_*, there was electrostatic force (*F_el_*) in Figure 12c between the heterostructure and the bottom silicon substrate, resulting in strain in Cr_2_Ge_2_Te_6_ thin films. The strain of the CGT layer can be estimated using the following formula:(3)$$\begin{array}{cc} \Delta \epsilon & =\epsilon_{el}\left(V_{g}\right)+\epsilon_{b}\left(V_{g}\right)\\ & \approx \left[{\frac{2}{3}\left(\frac{\varepsilon_{0}r}{8{g_{0}}^{2}n\left(T_{0}\right)}\right)}^{2}\right]V_{g}^{4}+\left[\frac{\varepsilon_{0}r∆z\left(1-v\right)}{4{g_{0}}^{2}n\left(T_{0}\right)}\right]V_{g}^{2}, \end{array}$$

Note that Δϵ is the total strain in the CGT layer; $\epsilon_{el}$ is the electrostatic pulling strain; $\epsilon_{b}$ is bending strain; $\varepsilon_{0}$ is the dielectric constant of vacuum; $V_{g}$ is the gate voltage; r is the membrane radius; v is the Poisson’s ratio; $n\left(T_{0}\right)$ is total tension in suspended heterostructures at $T_{0}=60\;K$; and $g_{0}$ is the separation between the heterostructure and the bottom Si substrate. Interestingly, when the electrostatic force induced a strain of 0.026%, the *T_C_* of the suspended Cr_2_Ge_2_Te_6_ /WSe_2_ heterostructure was increased by about 2.6 K, as shown in Figure 12d–g.

### 3.5. Field-Cooling

Due to the fact that the lattice parameters of CGT materials at 5 K are greater than those at 270 K, it would cause in-plane expansion during cooling [133,134]. Phatak et al. [45] found that field-cooling could cause the material expansion in CGT flakes, leading to in-plane strain. Furthermore, they directly observed strain-induced evolution of the magnetic domain structure by cryogenic Lorentz transmission electron microscopy (LTEM), which was closely related to the magnetoelastic coupling between strain and magnetization. This work revealed how to directly measure magnetic domain structures at the nanoscale.

## 4. Conclusions and Outlook

In this review, we have summarized the recent progress of strain-mediated intrinsic FM in 2D van der Waals materials with long-range order. First, we introduce how to explain the strain-mediated intrinsic FM on Cr-based and Fe-based 2D van der Waals materials with long-range FM order through ab initio Density functional theory (DFT), and how to calculate magnetic anisotropy energy (MAE) and T_C_ from the interlayer exchange coupling. Subsequently, we focus on numerous attempts to apply strain to 2D materials in experiments, including wrinkle-induced strain, flexible substrate bending or stretching, lattice mismatch, electrostatic force and field-cooling. However, research in this field is still in its early stages and there are many challenges that must be overcome.

By selecting substrates with different thermal expansion coefficients, including x-quartz [135,136,137,138], sapphire [27,28,139], and SiO_2_ [114]), uniaxial [135,136,137,138] or biaxial strain [27,28,114,139,140] could be introduced into 2D materials. The introduction of biaxial strain in this way was limited to the study of 2D materials without intrinsic long-range magnetic order, such as MoS_2_ [28] and ReS_2_ [27]. So far, this strategy has not been applied to the research on 2D intrinsic van der Waals materials. Very few equipment [47,48,121] could apply strain to 2D materials, but less could be coupled with magnetic testing equipment [41,141]. Especially, it is very challenging that one could accurately determine the strain while measuring the magnetic properties and T_C_ in different regions.

Recently, some emerging magnetic imaging technologies, such as magneto-optical Kerr effect (MOKE) [7,8,40,142,143,144], magnetic circular dichroism (MCD) [62,126,145,146,147,148,149], photoemission electron microscopy (PPEM) [150], scanning transmission X-ray microscopy (STXM) [151], Lorentz transmission electron microscopy (LTEM) [150,151,152], spin-polarized scanning tunneling microscopy (SP-TEM) [66,153], MFM [41,154], scanning SQUID [155,156,157,158,159], and scanning nitrogen-vacancy center microscopy (SNVM) [160,161,162,163,164], have been used to study 2D FM materials. However, these studies have not yet been combined with strain and, currently, it is almost impossible to directly analyze the relationship between strain and FM in 2D intrinsic materials. More importantly, strengthening the guideline of strain-mediated FM will promote the development of spintronics and straintronics.

## Figures and Tables

**Figure 1 nanomaterials-13-02378-f001:**
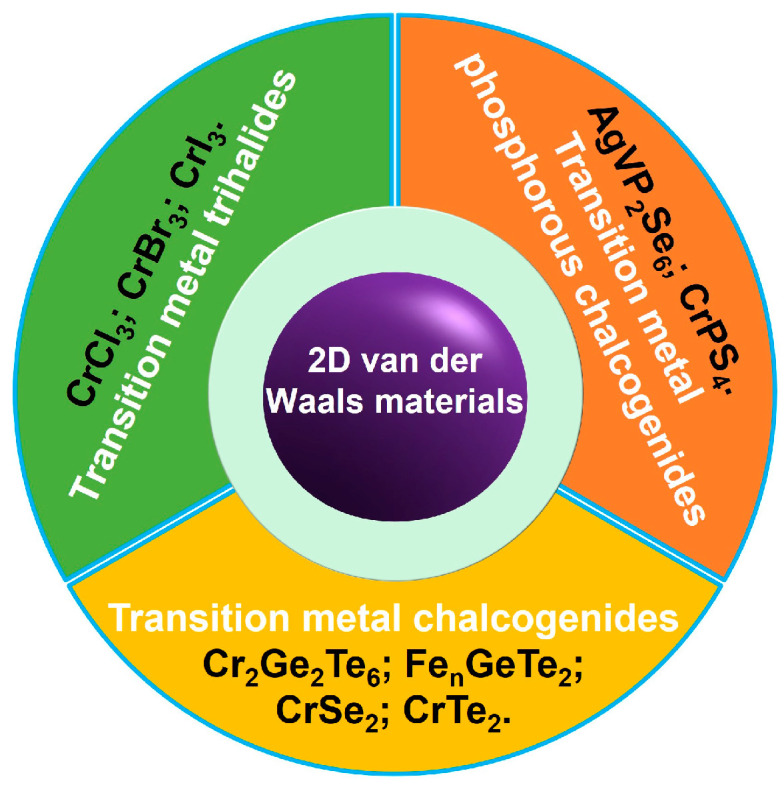
The three different kinds of 2D van der Waals materials with intrinsic long-range FM order.

**Figure 2 nanomaterials-13-02378-f002:**
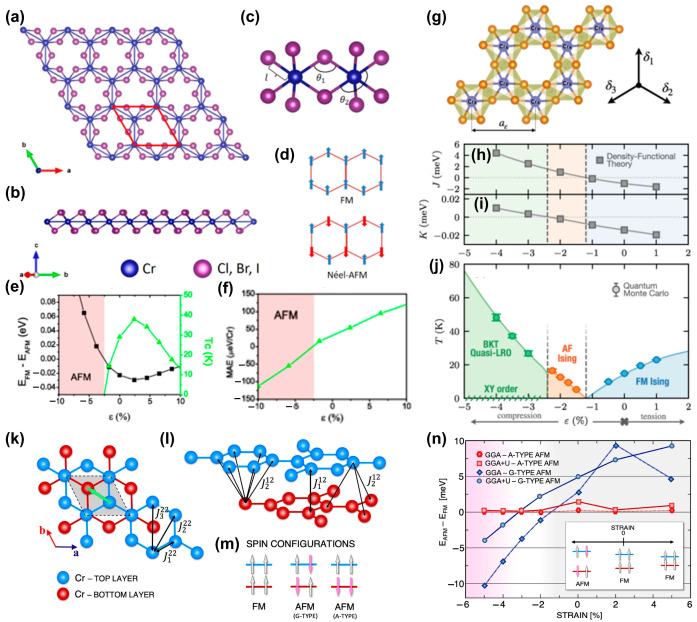
(**a**) Top and (**b**) side view of atomic structure of monolayer CrX_3_ (X = Cl, Br, I). (**c**) Bonding between chromium and iodine atoms. The unit cell of CrX_3_, which includes two Cr and six X atoms, has been indicated in (**a**). The bond length between Cr and an X atom, the bond angle *θ*_1_ between Cr and two X atoms in the same plane, and the axial angle *θ_2_* are also shown in (**c**). (**d**) The two magnetic orders, namely AFM and FM. Energy difference between the FM and AFM phases for (**e**) CrCl_3_. The AFM phase region is highlighted in red. The calculated *T_C_* is also shown for each case. Change in MAE with respect to strain in (**f**) CrCl_3_. (Reproduced with permission from [59]. Copyright 2016, American Physical Society). (**g**) Crystal structure of monolayer CrCl_3_. Dashed lines denote the unit cell with basic vectors $\delta_{1}=a_{\varepsilon }\left(0,1\right),\delta_{2}=a_{\varepsilon }\left(\sqrt{3}/2,\;\;-1/2\right)$, $\delta_{3}=a_{\varepsilon }\left(-\sqrt{3}/2,-1/2\right),$ with strain-dependent lattice constanta $a_{\varepsilon }$. (**h**,**i**) Magnetic nearest-neighbor superexchange J and anisotropy *K* of Hamiltonian (1), respectively, computed via Density Functional Theory (DFT) as a function of monolayer strain ε. (**j**) Finite−temperature phase diagram of the monolayer CrCl_3_ versus strain ε. Strain drives the monolayer into three different finite−temperature magnetic phases: BKT quasi-LRO phase, AFM Ising, and FM Ising. (Reproduced with permission from [63]. Copyright 2021, American Physical Society). (**k**,**l**) Arrangement of Cr atoms in the lattice of the CrCl_3_ bilayer in the low temperature phase. The green arrow indicates the direction of lateral shift between the top and bottom layers. (**m**) Schematic plot of three different spin configurations of bilayer CrCl_3_. (**n**) The energy difference between the FM and AFM phases. (Reproduced with permission from [64]. Copyright 2023, Springer Nature).

**Figure 3 nanomaterials-13-02378-f003:**
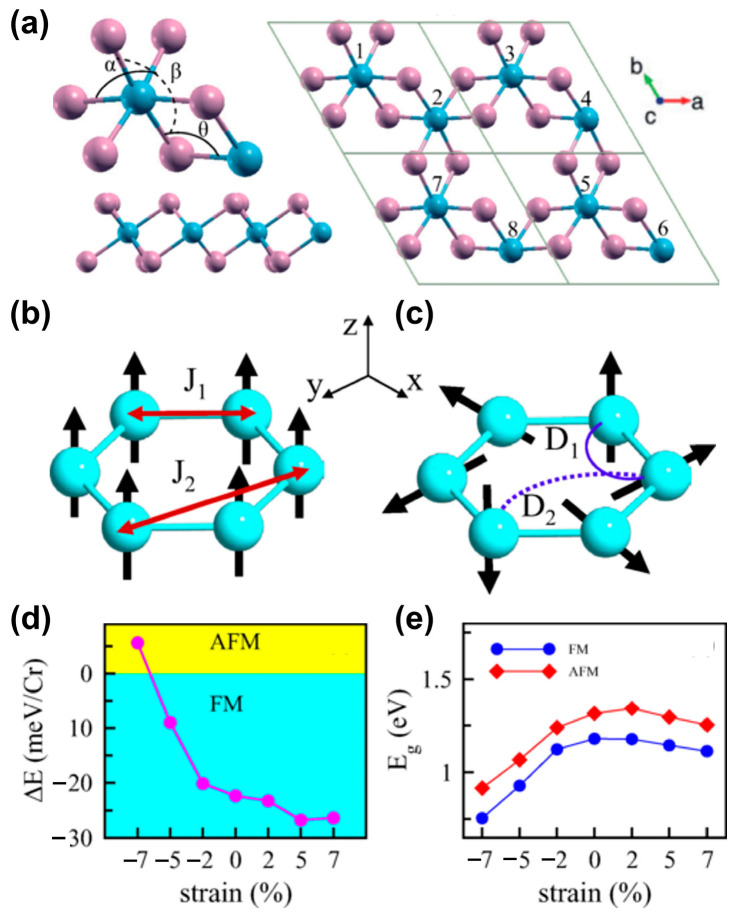
(**a**) Top and side views of monolayer CrI_3_. The bonding angles between the atoms in the monolayer are denoted by θ, α, and β. A 2 × 2 × 1 supercell of the monolayer. The blue (pink) spheres represent Cr(I) atoms. (**b**) Schematic picture of the symmetric exchange couplings between Cr atoms, where J_1_ denotes the coupling between nearest-neighbor atoms and J_2_ denotes the coupling between next-nearest-neighbor atoms. (**c**) The same as (**b**) for the DM vectors. (**d**) The total energy difference between the FM and AFM configurations, ΔE, and (**e**) the variation in the band gap as a function of strain for monolayer CrI_3_ in the FM and AFM configurations. (Reproduced with permission from [71]. Copyright 2020, American Physical Society).

**Figure 4 nanomaterials-13-02378-f004:**
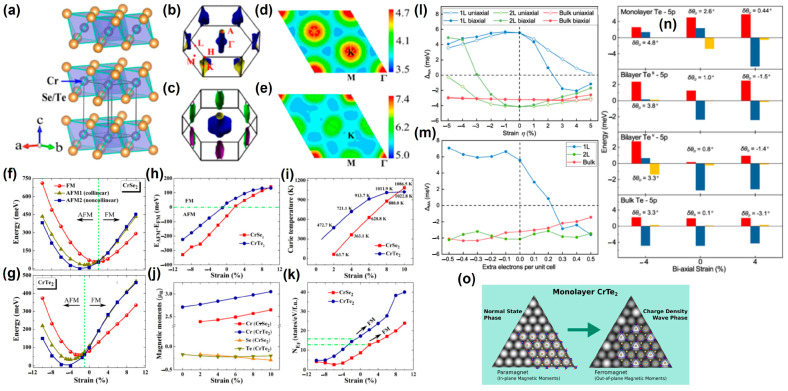
(**a**) Crystal structure of CrSe_2_/CrTe_2_ bulk; Fermi surface without spin polarization for (**b**) CrSe_2_ and (**c**) CrTe_2_ bulks; real part of electron susceptibility $\chi^{\prime }$ with $q_{z}=0$ for (**d**) CrSe_2_ and (**e**) CrTe_2_ bulks. Relative total energies of three different magnetic configurations [FM, AFM1 (collinear), and AFM2 (non-collinear)] as a function of biaxial strain for (**f**) CrSe_2_ and (**g**) CrTe_2_ monolayers. Strain dependence of (**h**) the energy difference ΔE (=*E_AFM_* − *E_FM_*) between AFM and FM states in one unit cell, (**i**) the *T_C_* in FM states, (**j**) magnetic moments on Cr and Se/Te atoms in FM states, and (**k**) the number of density of states (DOS) at the Fermi energy *N_EF_* in the nonmagnetic (NM) states for CrSe_2_/CrTe_2_ monolayers. (Reproduced with permission from [105]. Copyright 2015, American Physical Society). (**l**) strain and (**m**) band filling of 1 L, 2 L, and bulk 1T-CrTe_2_. Difference in SOC matrix elements Δ *_<pi|pj>_* (per atom) of the Te-5p orbitals. (**n**) Δ *_<pi|pj>_* of 1 L, 2 L, and bulk 1T-CrTe_2_ versus strain. At each strain, the values for *δθ_D_* are also shown. Positive and negative values of the strain correspond to tensile and compressive strain, respectively. For the bilayer, values for Te atoms at the van der Waals gap (Te^v^) and Te atoms at the free surface Te^s^ are shown. (Reproduced with permission from [106]. Copyright 2022, American Physical Society). (**o**) Computed STM images for monolayer CrTe_2_ in the normal state and the charge density wave (CDW) phase. (Reproduced with permission from [107]. Copyright 2020, American Chemical Society).

**Figure 5 nanomaterials-13-02378-f005:**
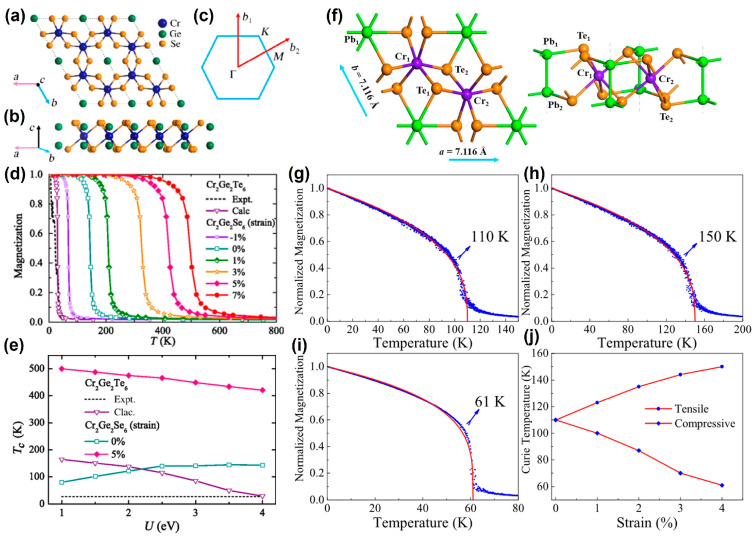
(**a**) top view in the *a*-*b* plane and (**b**) side view in the *a*-*c* plane of crystal structure of Cr_2_Ge_2_Se_6_. The 2D Brillouin zone is shown in (**c**). (**d**) *M*-*T*. The experimental results for are taken from Ref. [7]. The calculated results are obtained by DFT calculations and Monte Carlo simulations. (**e**) *T_C_*-*U*. (Reproduced with permission from [75]. Copyright 2019, American Physical Society). (**f**) top and side view of crystal structure of CrPbTe_3_. *M*-*T* curves for (**g**) pristine CrPbTe_3_, (**h**) 4% tensile strain structure, (**i**) 4% compressive strain structure and (**j**) variation of *T_C_* with strain. (Reproduced with permission from [78]. Copyright 2020, IOP Publishing).

**Figure 6 nanomaterials-13-02378-f006:**
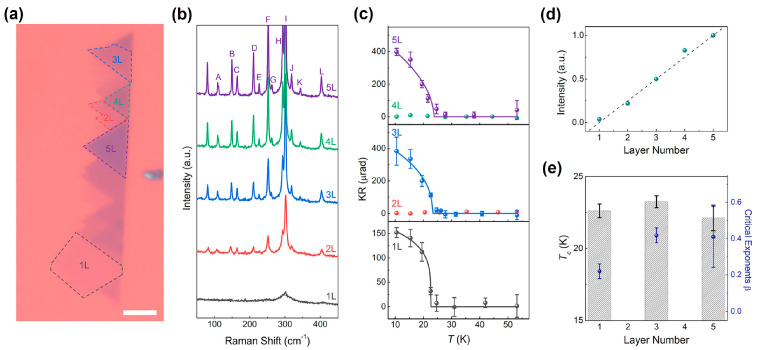
Layer dependent magnetic properties of thin CrPS_4_. (**a**) Optical image. Scale bar is 5 μm. (**b**) Raman spectra. (**c**) Temperature dependence of MOKE signal under μ_0_H = 0.25 T measured on 1 L to 5 L flakes shown in (**a**). (**d**) Raman peak intensity of the F peak for each flake normalized by that from the 5 L as a function of the layer number. (**e**) Critical temperature, *T_C_* (black dashed bar), and critical exponent, β (blue symbol), extracted from the fitting lines shown in (**c**). (Reproduced with permission from [68]. Copyright 2021, American Chemical Society).

**Figure 7 nanomaterials-13-02378-f007:**
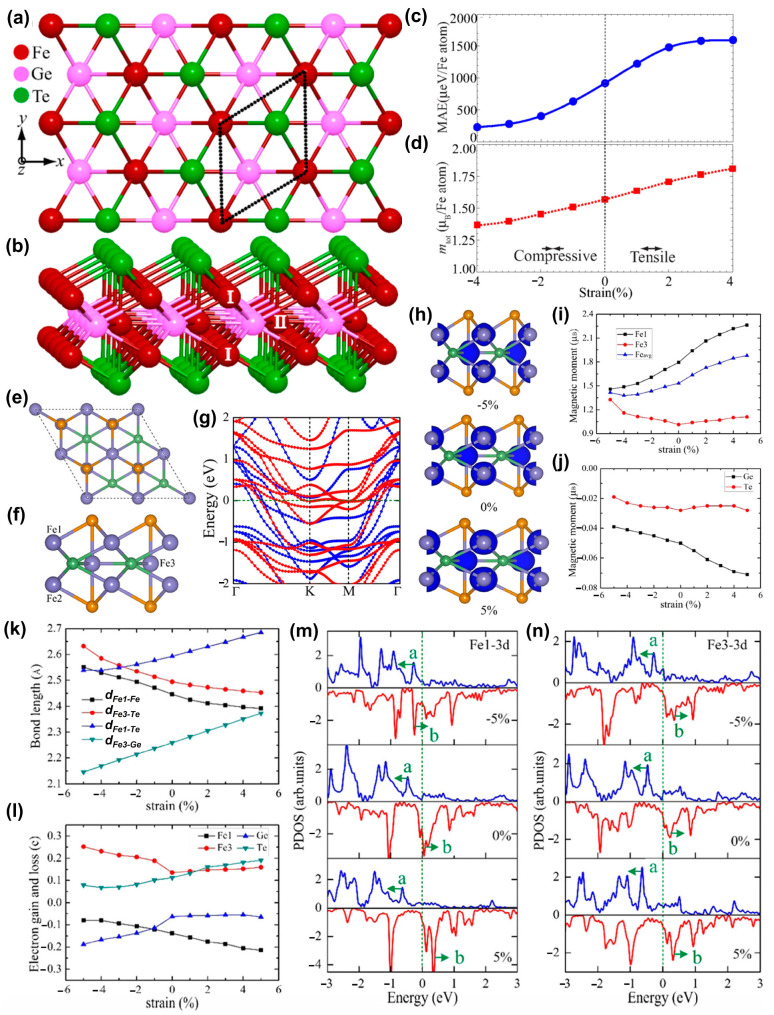
(**a**) Top and (**b**) side views of the atomic structure of monolayer Fe_3_GeTe_2_. The unit cell is enclosed by the dotted lines. Inequivalent Fe sites are numbered by I and II, respectively. Variation of (**c**) MAE and (**d**) total magnetic moment per Fe atom of single-layer Fe_3_GeTe_2_ under biaxial strain. (Reproduced with permission from [82]. Copyright 2016, American Physical Society). (**e**) Top and (**f**) side views of the structural models of the FGT monolayer. The purple, green, and yellow balls stand for Fe, Ge, and Te atoms, respectively. Fe sites are numbered by Fe1, Fe2, and Fe3, respectively, where Fe1 and Fe2 atoms are located at 2 equivsites, while Fe3 atom has the inequivalent site. (**g**) Spin-polarized band structures of the FGT monolayer. The Fermi level is set at zero, denoted by the olive dashed line. (**h**) Spin-density distribution of the FGT monolayer with −5, 0, and 5% strain. The isovalues are 0.02 e/Å^3^. Strain dependence of magnetic moment (**i**) per Fe1 and Fe3 atoms and (**j**) per Ge and Te atoms in the FGT monolayer. Strain dependence of (**k**) the distance and the bonding length (Fe1–Fe2 distance, *d_Fe1-Fe2_*; Fe3–Te bond length, *d_Fe3-Te_*; Fe1−Te bond length, *d_Fe1-Te_*; Fe3–Ge bond length, *d_Fe3-Ge_*). (**l**) Electron transfer of Fe, Ge, and Te atoms in the FGT monolayer. PDOS (partial density of states) of (**m**) Fe1 atom and (**n**) Fe3 atom in the FGT monolayer. The enhanced spin splitting of the a and b states in PDOS. (Reproduced with permission from [85]. Copyright 2020, American Chemical Society).

**Figure 8 nanomaterials-13-02378-f008:**
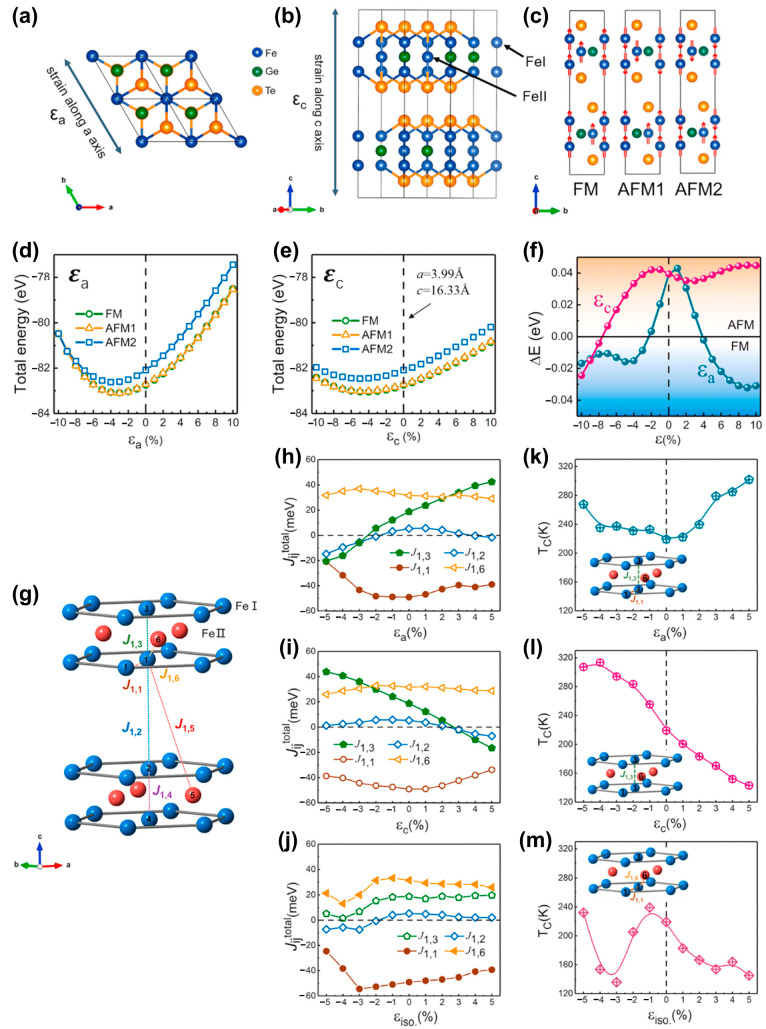
Schematic of the crystal structure of FGT from (**a**) top and (**b**) side view, and (**c**) the spin structure of FM, AFM1, and AFM2 configurations. (**d**,**e**) Total energy of FM, AFM1, and AFM2 configurations of FGT as a function of lattice distortion (**d**) along the a-axis and (**e**) the c-axis. (**f**) Total energy difference (Δ*E* = *E_FM_* − *E_AFM_*) under c-axis and a-axis strain. The background indicates the FM- or AFM-stable region. (**g**) The schematic picture of Fe–Fe exchange interactions in FGT, where only the Fe atoms are displayed. (**h**–**j**) Total isotropic exchange coupling parameters of J_1,1_, J_1,2_, J_1,3_, J_1,6_ in functions of three different kinds of distortion: (**h**) along the a-axis, (**i**) along the c-axis and (**j**) isotropically. (**k**–**m**) Corresponding variation of *T_C_* under above strains. The insets show the J*_i,_*_j_ that play a main role in the variation of *T_C_*. (Reproduced with permission from [86]. Copyright 2018, Elsevier).

**Figure 9 nanomaterials-13-02378-f009:**
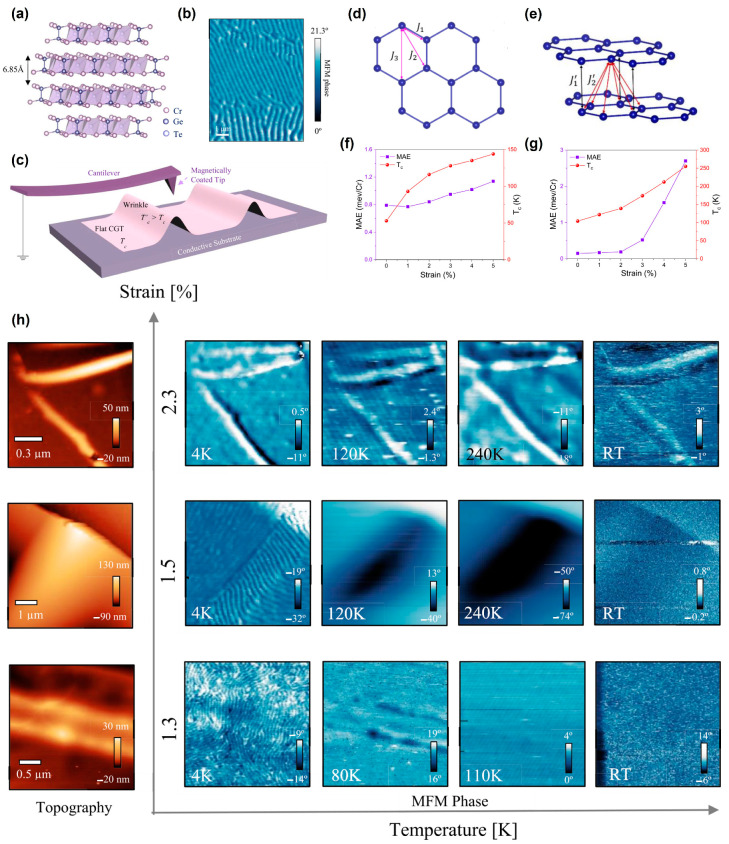
Curved nanostructures in *vdW* Cr_2_Ge_2_T_6_: (**a**) CGT crystal lattice showing the *vdW* layered structure and interlayer distance. (**b**) Typical magnetic stripe domain structure at 4 K seen in MFM measurements. (**c**) Schematic showing wrinkles in layered CGT and magnetically coated tip for MFM measurements. DFT predicted trend of magnetism and transition temperature with strain percentage. (**d**) Intralayer Cr the nearest-neighbor (J_1_), the second-nearest−neighbor (J_2_), and the third-nearest-neighbor (J_3_) exchange couplings. (**e**) Interlayer Cr the nearest−neighbor (J_1′_) and the second-nearest-neighbor (J_2′_) exchange couplings in bilayer Cr_2_Ge_2_T_6_. The calculated MAE per Cr atom and *T_C_* as functions of strain for (**f**) monolayer and (**g**) bilayer Cr_2_Ge_2_T_6_. (**h**) Temperature-dependent MFM examination of curved wrinkles with increasing strain: Left, Topography of three different strained wrinkles. Right, MFM image series depicting enhanced magnetic signal at the wrinkles depending on specific strain state up to RT. Magnetic signals in wrinkles exhibiting strain of 1.3% disappear above 100 K, while wrinkles with 2.3% strain exhibit clear MFM phase signals up to RT. Diagonal periodic lines in the figures are a result of instrument noise and are not part of the magnetic signal (1.3% strain at 110 K and 2.3% strain at 120 K). (Reproduced with permission from [42]. Copyright 2022, American Chemical Society).

**Figure 10 nanomaterials-13-02378-f010:**
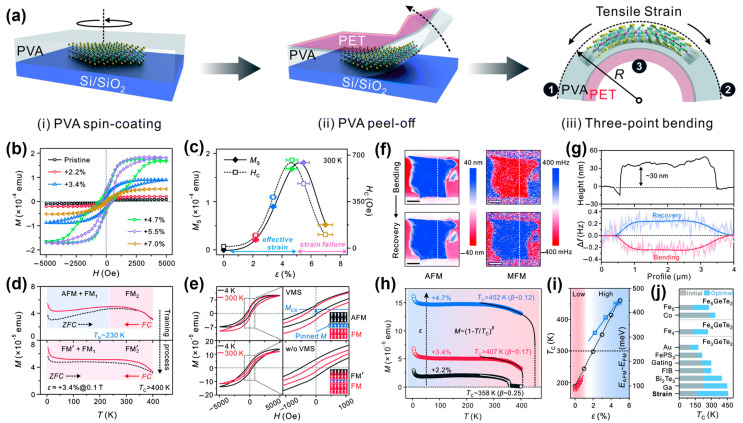
(**a**) Schematic illustration of the transfer and bending process of FGT nanoflakes on flexible PVA/PET substrates. (**b**) *M*–*H* curves at under different strain values. (**c**) Extracted *M_S_* and *H_C_* as a function of tensile strain. (**d**) ZFC and FC curves before and after the training procedure. (**e**) *M*–*H* curves and anomalous exchange-bias effect. Insets are the corresponding schematic illustrations. (**f**) AFM and MFM images of a single nanoflake in the bending and recovery states. All scale bars are 1 mm. (**g**) Extracted height and Df along the white dashed lines marked in (**f**). (**h**) *M*-*T* and critical fittings under different strains. (**i**) Strain-tuned *T_C_* phase diagram. The blue square symbols are the results of this current work and others are collected from the literature. (**j**) Comparison of optimal *T_C_* values using different methods based on the FGT system. From top to bottom: Fe_5_ [111] and Co [123] represent pure and Co-doped Fe_5_GeTe_2_, respectively; Fe_4_ refers to Fe_4_GeTe_2_ [124]; Au [125], FePS_3_ [37] and Bi_2_Te_3_ [34] denote the corresponding heterojunctions with Fe_3_GeTe_2_. Gating means the electrostatically gated Fe_3_GeTe_2_ [126], FIB [127] and Ga [128] are focused ion beam-treated and Ga ion-implanted Fe_3_GeTe_2_, respectively. (Reproduced with permission from [121]. Copyright 2021, The Royal Society of Chemistry).

**Figure 11 nanomaterials-13-02378-f011:**
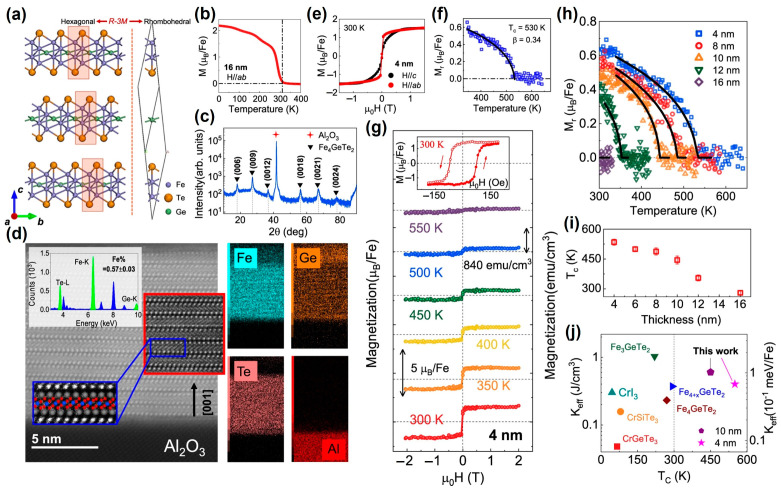
Crystal structure of the Fe_4_GeTe_2_ and its thickness-dependent magnetism. (**a**) Schematics of the crystal structure of Fe_4_GeTe_2_ stacked in ABC configuration (**left**), and its rhombohedral structure unit (**right**). (**b**) Temperature dependence of the magnetization for 16 nm Fe_4_GeTe_2_. (**c**) XRD scan of the Fe_4_GeTe_2_ film. (**d**) A typical HRTEM image of Fe_4_GeTe_2_ films; the color squares show the high-pass filtered images of the vdW structure. Up Inset: The EDX result verifies the 4:1:2 Fe: Ge: Te stoichiometric composition with the uniform element distribution map (right). I *M*–*H* curves of 4 nm Fe_4_GeTe_2_ at 300 K. (**e**) Room-temperature magnetic hysteresis loops. (**f**) In-plane *M_r_*–*T* curve. (**g**) Detailed magnetic field-dependent magnetization of 4 nm Fe_4_GeTe_2_ at various temperatures for *H_//ab_*. Inset: zoom-in hysteresis loop at 300 K. (**h**) *M_r_*-*T* curves. (**i**) *Tc* for Fe_4_GeTe_2_ thin films with different thicknesses. (**j**) Effective magnetic an isotropy energy *K_eff_* and curie temperature *Tc* for our samples and previous vdW ferromagnets [77,124,129,130,131]. (Reproduced with permission from [43]. Copyright 2023, Springer Nature).

**Figure 12 nanomaterials-13-02378-f012:**
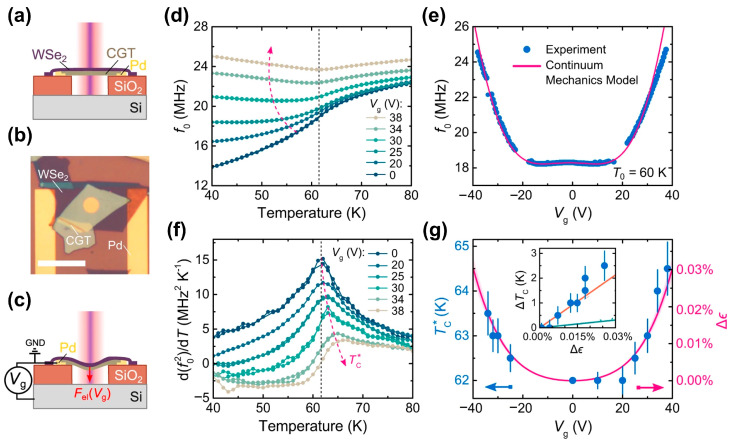
Strain-mediated *T_C_* in a suspended CGT/WSe_2_ heterostructure. (**a**) The schematic cross-section of the suspended CGT/WSe_2_ heterostructure membrane. (**b**) The optical image of the heterostructure. Scale bar: 12 μm. (**c**) The schematics of the electrostatic strain-tuning principle. (**d**) Measured resonance frequencies *f_0_* of the heterostructure membrane as a function of temperature for different gate voltages *V_g_*. (**e**) Filled blue circles—the measured resonance frequency as a function of *V_g_* at 60 K. (**f**) The temperature derivative of *f_0_*^2^ as a function of temperature. (**g**) Solid magenta line—the estimate of electrostatically induced strain Δϵ as a function of *V_g_*. The shaded magenta region shows the uncertainty in Δϵ. The inset shows as a function of added voltage-induced strain Δϵ in addition to calculations from Li and Yang [73] (solid green line) and from Dong et al. [132] (solid orange line). The vertical error bars in *T_C_* were estimated from determining the peak position in (**e**) within 2% accuracy in the measured maximum. (Reproduced with permission from [44]. Copyright 2022, American Chemical Society).

## Data Availability

Not applicable.

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
