# Peer review of "Strain Engineering of Intrinsic Ferromagnetism in 2D van der Waals Materials"

_nanomaterials, 2023, doi:10.3390/nano13162378_

Round 1

Reviewer 1 Report

Nanomaterials_July19-2023Referee report

Nanomaterials

Manuscript ID: nanomaterials-2522970

Strain Engineering of Intrinsic Ferromagnetism in 2D van der Waals Materials

Hongtao Ren, Gang Xiang

The manuscript is devoted to the survey of the latest results in the field of strain engineering of intrinsic ferromagnetism in 2D vdW materials. The recent progress of strain engineering of intrinsic ferromagnetism low-dimensional materials has been summarized, namely: i) theoretical progress of FM strain tuning; ii) experimental results of strain-mediated FM in 2D materials using various techniques; and iii) electrostatic force and field-cooling. Several type of magnetic 2D FM materials have been considered, namely i) transition metal trihalides like CrCl3, CrBr3 and CrI3; ii) transition metals chalcogenides (Cr2Ge2Te6, FenGeTe2, 46 CrTe2); and iii) transition metal phosphorous chalcogenides (AgVP2Se6, CrPS4). Based on the extended review of modern literature it was shown that despite many attempts with diverse approaches, the study of strain-dependent intrinsic FM in 2D materials is still in early stages, and a few potential directions have been proposed. Overall, a deeper and broader understanding of strain-mediated FM is very necessary, which may greatly affect the development the key spin-related applications.

The Review is clearly written, the contemporary field of strain manipulation of magnetic properties is well described and the hints for development of the field has been properly introduced. The text is written in clear English language. I’m pretty sure the Review will be a well-cited publication. I strongly recommend the Editorial Board of Nanomaterials to accept and publish the Review. 

--------------

Disclaimer: All publications citated in this referee report are of fundamental importance for this review. The reviewer neither confirms or denies the authorship of any cited publication and does not require to cite any of them in revised version of the manuscript.

Author Response

We appreciate the reviewer’s evaluation and all of the comments she/he raised during the review process.

Reviewer 2 Report

Dear Editors,

This is a very nice review about the possible existence of a room-temperature ferromagnetic structure in 2D of the vdW type. Several materials have been reviewed and experimental results have been complemented by theoretical calculations. The authors focused their attention on the possibility of tuning the magnetic material properties through strain engineering. In my opinion, this is a great contribution to the field and the Review can be published as it is.

Author Response

(The authors gave the same response as above.)

Reviewer 3 Report

2D magnetic materials are at the cutting edge of contemporary materials science. Controlling their magnetism by strain constitutes a promising route to tuning their properties to maximize applicability is spintronics (including the influence on the critical temperature). The review, therefore, focuses on important subject of interest to the community. It is quite comprehensive and presents a useful perspective on the emerging field. I recommend the manuscript for publication in Nanomaterials, provided that the Authors give prior consideration to the points listed below, mainly related to some presentation details.

* Adding a list of all abbreviations at the end of the paper would be useful to the Readers.

* Introduction: the proximity-induced magnetism would be mentioned as some alternative to intrinsic one (with some examples) to give some wider context to the considerations.

* Line 29: “in 2004” should replace “on 2004”.

* Line 51, the beginning of section "Theoretical calculations": some introduction to open the section would be added - for example focusing on the methods used for calculations (ab initio DFT approach) as well as methods used to evaluate the critical temperature from the knowledge of exchange integrals. I believe that, in order to strengthen somehow the theoretical background regarding intrinsic magnetism in 2D materials, such works as https://doi.org/10.1021/acsnano.1c09150 and https://doi.org/10.1088/2053-1583/aaf06d might be cited to enrich the discussion. 

* Line 59: “Curie” instead of “curie”.

* Line 61: “out of plane” might sound better than “off-plane”.

* Figure 1: “waals” should be replaced with “Waals”. Moreover, “intrinsic magnetic materials” would be more clear.

* Line 67, eq. (1): the symbol „k” should be replaced with “K” to be in concert with the axis label in Fig. 2(i). Moreover “(S)^z” should replace with “(S)^2” (square of spin operator). In addition, I guess that in the first sum, vector arrows should be placed over the spin operators S, as it is Heisenberg Hamiltonian in original paper [39]. Also the spin value of S=3/2 should be mentioned there.

* Line 69: “BKT quasi-LRO phase” abbreviations should be explained.

* Line 76: “Tc” should be explained somewhere as Curie temperature.

* Line 76/77: the kind of doping mentioned there should be explained (after Ref. 70).

* Line 105, 107 and 116: “electronic bandgap” might be more correct.

* Line 111: “spin-relevant” semiconductor sounds unclear and it should be explained or rephrased.

* Line 113: “if”->”of”.

* Line 114: “similar phase transition” would be explained, as it is not clear to what transition it is similar.

* Line 118: DM should be explained as Dzialoshinskii-Moriya.

* Figure 3(b) the figure panel contains explanation of both symmetric coupling J and relevant vectors for Dzialoshinskii-Moriya interactions, whereas the caption of panel (b) refers only to the former interaction. This should be supplemented.

* Line 159: “a” should replace “s”.

* Line 165: probably “ferromagnet” would sound better than “ferromagnetism”.

* Line 195: “related” instead of “relate”.

* Figure 6: as some of the panels present dependences of exchange integrals between particular spin pairs, a schematic view of the structure with the positions marked would be valuable-if it is present in original work.

* Lines 220-223: Tc should be written as mathematical symbol.

* Line 227: “d” in italics.

* Line 251: “strain-free” would sound better.

* Line 253: The term “increase of the density of states” would be enough.

* Line 257: the sentence about bandgap filling would be rephrased, as it is unclear.

* Line 279: The title “Modifications” would be modified to better reflect the content of the section.

* Line 287: the sentence “magnetic moment of the flat flake was nonmagnetic” does not sound good.

* Line 291: PAD would be explained.

* Line 321-322: the sentence “the hysteresis loops exhibited soft or hard FM states.” would be extended for more clarity.

* Line 322: “those found for the case of tensile strain” would sound better.

* Line 383: r in M_r should be in lower index.

* Line 421-2: the sentence “Because the lattice constant of CGT material is larger at low temperature, it will cause in-plane expansion” might be extended to be more clear.

* References: in some cases “van der waals” would be changed to “van der Waals” in the titles.

The detailed remarks - see section above.

Author Response

Reviewer: 3

2D magnetic materials are at the cutting edge of contemporary materials science. Controlling their magnetism by strain constitutes a promising route to tuning their properties to maximize applicability is spintronics (including the influence on the critical temperature). The review, therefore, focuses on important subject of interest to the community. It is quite comprehensive and presents a useful perspective on the emerging field. I recommend the manuscript for publication in Nanomaterials, provided that the Authors give prior consideration to the points listed below, mainly related to some presentation details.

[Comment]:

Point 1: Adding a list of all abbreviations at the end of the paper would be useful to the Readers.

[Response]:

Thank you very much for taking time to review our manuscript and the positive comments on our manuscript. In response to your point 1, we have added the list of all abbreviations in our revised manuscript on page 15.

Abbreviations:

2D                              two-dimensional

3D                              three-dimensional

AFM                            antiferromagnetism

BKT                             Berezinskii-Kosterlitz-Thouless

CDW                           charge density wave

CGT                         Cr2Ge2Te6

CPT                             CrPbTe3

DFT                            density functional theory

DM                             Dzialoshinskii-Moriya

DOS                            density of states

FGT                         Fe3GeTe2

FM                             ferromagnetism

GGA                        the generalized gradient approach

HRTEM                      high resolution transmission electron microscopy

LDA                         linear density approach

LRO                            long-range order

LSDA                       local spin density approximation

LTEM                        lorentz transmission electron microscopy

MAE                           magnetic anisotropy energy

MBE                            molecular beam epitaxy

MC                          Monte Carlo

MCD                         magnetic circular dichroism

MFM                           magnetic force microscopy

MOKE                       magneto-optical Kerr effect

NM                             nonmagnetism

PAD                            polymer assisted deposition

PDMS                       polydimethylsiloxane

PEEM                        photoemission electron microscopy

PET                         polyethyleneterephthalate

PI                           polyimide

PRA                         the random phase approximation

PVA                         polyvinyl alcohol

QMC                           Quantum Monte Carlo

RT                           room temperature

SEM                         scanning electron microscopy

SNVM                       scanning nitrogen-vacancy center microscopy

SQUID                      superconducting quantum interference device magnetometry

SOC                            spin-orbit coupling

SP-TEM                      spin-polarized scanning tunneling microscopy

SSR                             solid-state reaction

STM                            scanning tunneling microscopy

STXM                           scanning transmission X-ray microscopy

XRD                         X-ray diffraction

Fel                             electrostatic force

TC                              curie temperature

J                                the exchange coupling constant

K                               magnetic anisotropy

                              total energy difference[Comment]:
Point 2: Introduction: the proximity-induced magnetism would be mentioned as some alternative to intrinsic one (with some examples) to give some wider context to the considerations.

[Response]:

Thank you very much for taking time to review our manuscript and the positive comments on our manuscript. In response to your point 2, we have added the relevant sentences on Line 44-46.

“More interestingly, the Bi2Te3|Fe3GeTe2 heterostructure related to strain [29, 30], have been designed to increase the Curie temperature (TC) due to the proximity effect [31-35].”

[Comment]:

Point 3: Line 29: “in 2004” should replace “on 2004”.

[Response]:

Thank you very much for your comment. According to your suggestion, we have substituted “in 2004” for “on 2004” on Line 38 in the revised manuscript.

[Comment]:

Point 4: Line 51, the beginning of section "Theoretical calculations": some introduction to open the section would be added - for example focusing on the methods used for calculations (ab initio DFT approach) as well as methods used to evaluate the critical temperature from the knowledge of exchange integrals. I believe that, in order to strengthen somehow the theoretical background regarding intrinsic magnetism in 2D materials, such works as https://doi.org/10.1021/acsnano.1c09 150 and https://doi.org/10.1088/2053-1583/aaf06d might be cited to enrich the discussion.

[Response]:

We thank the reviewer for this valuable suggestion. According to your suggestion, we have added the relevant description and the two references as Refs. 6 and 97 on page 2 in the revised manuscript.

“In order to understand the essence of 2D ferromagnetism, ab initio Density functional theory, including linear density approach (LDA) [59], local spin density approximation (LSDA) [92], the generalized gradient approach (GGA) [61, 93, 94], and DFT+U [95, 96] was often used to calculate the electronic structure of the system as a starting point. Moreover, the interlayer exchange coupling J was closely related to magnetic anisotropy, and it would also be used to calculate TC [86, 97-100]. The mean field theory [97, 101] would roughly estimate TC, but TC often was overestimated it. Although the random phase approximation (RPA) could more accurately estimate TC of three-dimensional (3D) materials, it may fail in 2D systems with large anisotropy. Notably, classic Monte Carlo (MC) [6, 97] simulations can also well describe the critical temperature.”

[Comment]:

Point 5: Line 59: “Curie” instead of “curie”;

Line 61: “out of plane” might sound better than “off-plane”.

[Response]:

Thank you very much for your comment. According to your suggestion, we have substituted “Curie” for “curie” (Line 84) and “off-plane” for “out of plane” (Line 86) in the revised manuscript.

[Comment]:

Point 6: Figure 1: “waals” should be replaced with “Waals”. Moreover, “intrinsic magnetic materials” would be more clear.

[Response]:

Thank you very much for your comment. According to your suggestion, we have substituted “Waals” for “waals” on Figure 1 of the revised manuscript. In addition, we have revised the relevant description on Line 43-44, 51, 60, 314-315 and 470-471.

“2D van der Waals materials with intrinsic long-range FM order” on Line 43-44.

“2D materials with intrinsic long-range FM order” on Line 51.

“2D van der Waals materials with intrinsic long-range FM order” on Line 60.

“2D materials without intrinsic long-range magnetic order” on Line 314-315.

“2D materials without intrinsic long-range magnetic order” on Line 470-471.

[Comment]:

Point 7: Line 67, eq. (1): the symbol „k” should be replaced with “K” to be in concert with the axis label in Fig. 2(i). Moreover “(S)^z” should replace with “(S)^2” (square of spin operator). In addition, I guess that in the first sum, vector arrows should be placed over the spin operators S, as it is Heisenberg Hamiltonian in original paper [39]. Also the spin value of S=3/2 should be mentioned there.

[Response]:

Thank you very much for pointing out this mistake. Following your suggestion, we have revised the eq. (1) on Line 93 and added the relevant description 94 on Line in our manuscript.

“Note that: the spin value of S was 3/2 in the above equation.”

[Comment]:

Point 8: Line 69: “BKT quasi-LRO phase” abbreviations should be explained.

[Response]:

Thank you very much for your helpful suggestion. Following your suggestion, we have revised the sentence on Line 96 in our manuscript.

“BKT (Berezinskii-Kosterlitz-Thouless) quasi-long-range order (LRO) phase”

[Comment]:

Point 9: Line 76: “Tc” should be explained somewhere as Curie temperature.

[Response]:

Thank you very much for pointing out this mistake. Following your suggestion, we have revised the sentence on Line 124 in our manuscript.

[Comment]:

Point 10: Line 76/77: the kind of doping mentioned there should be explained (after Ref. 70).

[Response]:

Thank you very much for your helpful suggestion. Following your suggestion, we have added the relevant description on Line 125-127.

“Although both hole doping and electron doping could enhance ferromagnetic coupling, the effect of hole doping was better at the same doping concentration”

[Comment]:

Point 11: Line 105, 107 and 116: “electronic bandgap” might be more correct.

[Response]:

Thank you very much for pointing out this mistake. Following your suggestion, we have revised the sentence on Line 135, 137 and 146.

[Comment]:

Point 12: Line 111: “spin-relevant” semiconductor sounds unclear and it should be explained or rephrased.

[Response]:

Thank you very much for your helpful suggestion. Following your suggestion, we have substituted “magnetic” for “spin-relevant” on line 141 of the revised manuscript.

[Comment]:

Point 13: Line 113: “if”->“of”

[Response]:

Thank you very much for pointing out this mistake. We have corrected it on line 144 the revised manuscript.

[Comment]:

Point 14: Line 114: “similar phase transition” would be explained, as it is not clear to what transition it is similar.

[Response]:

Thank you very much for your comment. According to your suggestion, we have rephased it on Line 144-145 in our revised manuscript.

“A similar phase transition from FM to AFM”

[Comment]:

Point 15: Line 118: DM should be explained as Dzialoshinskii-Moriya.

[Response]:

Thank you very much for your comment. According to your suggestion, we have revised have revised the relevant description on Line 148 of the revised manuscript.

[Comment]:

Point 16: Figure 3(b) the figure panel contains explanation of both symmetric coupling J and relevant vectors for Dzialoshinskii-Moriya interactions, whereas the caption of panel (b) refers only to the former interaction. This should be supplemented.

[Response]:

Thank you very much for pointing out this mistake. We have added the relevant description in the caption of Fig. 3 on Line 169.

“(c) The same as (b) for the DM vectors.”

[Comment]:

Point 17: Line 159: “a” should replace “s”.

Line 165: probably “ferromagnet” would sound better than “ferromagnetism”.

Line 195: “related” instead of “relate”.

[Response]:

Thank you very much for your comment. According to your suggestion, we have substituted “a” for “s” on Line 239, “ferromagnet” for “ferromagnetism” on Line 245, and “related” for “relate” on Line 260 in the revised manuscript.

[Comment]:

Point 18: Figure 6: as some of the panels present dependences of exchange integrals between particular spin pairs, a schematic view of the structure with the positions marked would be valuable-if it is present in original work.

[Response]:

Thank you very much for your comment. According to your suggestion, we have added a schematic view of the structure and the relevant description on Figure 8 of the revised manuscript.

“(g) The schematic picture of Fe–Fe exchange interactions in FGT where only the Fe atoms are dis-played.” on page 306-307.

[Comment]:

Point 19: Lines 220-223: Tc should be written as mathematical symbol.

[Response]:

Thank you very much for your comment. According to your suggestion, we have rewritten  as mathematical symbol on Line 289 in the revised manuscript.

[Comment]:

Point 20: Line 227: “d” in italics.

Line 251: “strain-free” would sound better.

Line 253: The term “increase of the density of states” would be enough

[Response]:

Thank you very much for your comment. According to your suggestion, we have corrected it on Line 299, Line 160 and Line 162 in the revised manuscript.

[Comment]:

Point 21: Line 257: the sentence about bandgap filling would be rephrased, as it is unclear.

[Response]:

Thank you very much for your comment. According to your suggestion, we have substituted “band” for “bandgap” on Line 192 and cited a relevant reference as Refs. [108] in our revised manuscript.

[108] Yan, S.; Wang, X. C.; Mi, W. B. Role of electron filling in the magnetic anisotropy of monolayer WSe2 doped with 5d transition metals. Phys. Rev. Mater. 2017, 1, 074408.

[Comment]:

Point 22: Line 279: The title “Modifications” would be modified to better reflect the content of the section.

[Response]:

Thank you very much for your comment. According to your suggestion, we have revised the title on Line 312 in the revised manuscript.

“Introducing strain in 2D van der Waals materials”

[Comment]:

Point 23: Line 287: the sentence “magnetic moment of the flat flake was nonmagnetic” does not sound good.

        Line 291: PAD would be explained.

[Response]:

Thank you very much for your comment. In response to your comment and improve the storytelling of the paper, we have deleted and revised the relevant description at the section 3.1 on page 10-11.

[Comment]:

Point 24: Line 321-322: the sentence “the hysteresis loops exhibited soft or hard FM states.” would be extended for more clarity.

[Response]:

Thank you very much for your comment. According to your suggestion, we have rewritten it on Line 352-354 in our revised manuscript.

“When the strain reached 4.7%, the hysteresis loop evolved from soft magnet to hard magnet; However, when the strain reached 7.0%, the hysteresis loop displayed an opposite evolution trend.”

[Comment]:

Point 25: Line 322: “those found for the case of tensile strain” would sound better.

Line 383: r in Mr should be in lower index

[Response]:

Thank you very much for your comment. According to your suggestion, we have revised it on Line 355 and Line 412 of the revised manuscript.

[Comment]:

Point 26: Line 421-2: the sentence “Because the lattice constant of CGT material is larger at low temperature, it will cause in-plane expansion” might be extended to be more clear.

[Response]:

Thank you very much for your comment. According to your suggestion, we have rewritten it on Line 450-451 in our revised manuscript.

“Due to the fact that the lattice parameters of CGT materials at 5K are greater than those at 270K, it would cause in-plane expansion during cooling [133, 134]”

[Comment]:

Point 27: References: in some cases “van der waals” would be changed to “van der Waals” in the titles.

[Response]:

Thank you very much for your comment. According to your suggestion, we have revised it in the titles of References.

Reviewer 4 Report

The paper reports a review about strain engineering in 2D ferromagnets based on van der Waals materials, which is indeed a very hot topic and a very interesting research direction. This work shows the big picture of the field, but I feel it empty and undeveloped, as it neglects all the progress that has been performed in the understanding and implementation of strain in 2D ferromagnets. The same happens with the writing, which I found not very developed. A clear example of this is the abstract, which should be the most important and representative part of the paper. The abstract is constructed with a rewritten version of the author’s previous publication’s abstract (citation 17) and two paragraphs copy-pasted from the introduction of this submitted paper.

"Strain engineering is an important strategy for regulating material properties, including optoelectronic, electrocatalytic, and magnetic properties. Due to the ability of two-dimensional (2D) materials to withstand up to 20% strain, it is possible to control the properties of 2D intrinsic ferromagnetic materials through strain engineering."

 First, we introduce theoretical progress of strain tuning of intrinsic FM in 2D van der Waals materials. Then, we discuss experimental results of strain-mediated FM in 2D materials with various strain introducing techniques, such as wrinkle-induced strain, flexible substrate bending or stretching, lattice mismatch, electrostatic force and field-cooling.”

 “Despite many attempts with diverse approaches, the study of strain-dependent intrinsic FM in 2D materials is still in early stages, and a few potential directions are proposed. Overall, a deeper and broader understanding of strain-mediated FM is very necessary, which will further drive the development of spintronics.”

The redaction is not attractive and focuses in small and technical details like the Curie temperature being increased 5K with strain without any rationalization that shows the understanding of ferromagnetism and strain that the community has nowadays. About the section 3, I support the structure and the idea of explaining the main techniques to perform strain 2D materials, however the writing should be focused in rationalizing and explaining the techniques and their potential and limitations, supported by the experiments in bibliography. At the moment this section just rewrites the explanation of experiments that were already explained in their respective papers, and does not provide a significative rationalization. Neither extracts a perspective or a big picture that explains how all these experiments contributed to the field.

The design of the figures is also very poor; the Figures are enormous panels that don’t even enter in the margins of the manuscript. They can show between 4-7 times the same structure of the material, given the message of the figures was not rationalized properly.

Finally, another very important aspect to be careful about is the possible presence of auto plagiarism. At least the contents of the 3.1 version were already explained in their previous Nanomaterials (citation 18) and the authors present a rewritten version.

In the actual stage, I do not recommend the manuscript for publication. If the authors want to improve the manuscript I strongly recommend to investigate further the literature about CrI3. There is a lot of publications (below I suggest some) that can help the authors to have a more complete perspective about the actual understanding of strain in 2D ferromagnets, to build the first part of the paper. About the section 3, I strongly recommend to read some reviews and papers related to the latest experiments in strain of 2D materials, this can help to extract a big picture and improve the storytelling of this second part of the paper.

Banasree Sadhukhan, Anders Bergman, Yaroslav O Kvashnin, Johan Hellsvik, and Anna Delin. Spin-lattice couplings in two-dimensional cri 3 from first-principles computations. Physical Review B, 105(10):104418, 2022.

Esteras, D. L., Rybakov, A., Ruiz, A. M., & Baldoví, J. J. (2022). Magnon straintronics in the 2D van der Waals ferromagnet CrSBr from first-principles. Nano Letters22(21), 8771-8778.

Kashin, I. V., Mazurenko, V. V., Katsnelson, M. I., & Rudenko, A. N. (2020). Orbitally-resolved ferromagnetism of monolayer CrI3. 2D Materials7(2), 025036.

Soriano, D., Rudenko, A. N., Katsnelson, M. I., & Rösner, M. (2021). Environmental screening and ligand-field effects to magnetism in CrI3 monolayer. npj Computational Materials7(1), 162.

Kvashnin, Y. O., Bergman, A., Lichtenstein, A. I., & Katsnelson, M. I. (2020). Relativistic exchange interactions in Cr X 3 (X= Cl, Br, I) monolayers. Physical Review B102(11), 115162.

Author Response

Reviewer: 4

The paper reports a review about strain engineering in 2D ferromagnets based on van der Waals materials, which is indeed a very hot topic and a very interesting research direction. In the actual stage, I do not recommend the manuscript for publication. If the authors want to improve the manuscript. I strongly recommend to investigate further the literature about CrI3. There is a lot of publications (below I suggest some) that can help the authors to have a more complete perspective about the actual understanding of strain in 2D ferromagnets, to build the first part of the paper.

About the section 3, I strongly recommend to read some reviews and papers related to the latest experiments in strain of 2D materials, this can help to extract a big picture and improve the storytelling of this second part of the paper.

Banasree Sadhukhan, Anders Bergman, Yaroslav O Kvashnin, Johan Hellsvik, and Anna Delin. Spin-lattice couplings in two-dimensional CrI3 from first-principles computations. Physical Review B, 105(10):104418, 2022.

Esteras, D. L., Rybakov, A., Ruiz, A. M., & Baldoví, J. J. (2022). Magnon straintronics in the 2D van der Waals ferromagnet CrSBr from first-principles. Nano Letters, 22(21), 8771-8778.

Kashin, I. V., Mazurenko, V. V., Katsnelson, M. I., & Rudenko, A. N. (2020). Orbitally-resolved ferromagnetism of monolayer CrI3. 2D Materials, 7(2), 025036.

Soriano, D., Rudenko, A. N., Katsnelson, M. I., & Rösner, M. (2021). Environmental screening and ligand-field effects to magnetism in CrI3 monolayer. npj Computational Materials, 7(1), 162.

Kvashnin, Y. O., Bergman, A., Lichtenstein, A. I., & Katsnelson, M. I. (2020). Relativistic exchange interactions in CrX3 (X= Cl, Br, I) monolayers. Physical Review B, 102(11), 115162.

[Comment]:

Point 1: The paper reports a review about strain engineering in 2D ferromagnets based on van der Waals materials, which is indeed a very hot topic and a very interesting research direction. This work shows the big picture of the field, but I feel it empty and undeveloped, as it neglects all the progress that has been performed in the understanding and implementation of strain in 2D ferromagnets. The same happens with the writing, which I found not very developed. A clear example of this is the abstract, which should be the most important and representative part of the paper. The abstract is constructed with a rewritten version of the author’s previous publication’s abstract (citation 17) and two paragraphs copy-pasted from the introduction of this submitted paper.

"Strain engineering is an important strategy for regulating material properties, including optoelectronic, electrocatalytic, and magnetic properties. Due to the ability of two-dimensional (2D) materials to withstand up to 20% strain, it is possible to control the properties of 2D intrinsic ferromagnetic materials through strain engineering."

 “First, we introduce theoretical progress of strain tuning of intrinsic FM in 2D van der Waals materials. Then, we discuss experimental results of strain-mediated FM in 2D materials with various strain introducing techniques, such as wrinkle-induced strain, flexible substrate bending or stretching, lattice mismatch, electrostatic force and field-cooling.”

 “Despite many attempts with diverse approaches, the study of strain-dependent intrinsic FM in 2D materials is still in early stages, and a few potential directions are proposed. Overall, a deeper and broader understanding of strain-mediated FM is very necessary, which will further drive the development of spintronics.”

Finally, another very important aspect to be careful about is the possible presence of auto plagiarism.

[Response]:

Thank you very much for your comment. According to your suggestion, we have rewritten the abstract in the revised manuscript.

“Since the discovery of low-temperature long-range ferromagnetic order in monolayer Cr2Ge2Te6 and CrI3, many efforts have been made to achieve a room temperature (RT) ferromagnet. Due to the outstanding deformation ability of two-dimensional (2D) materials, which provides an exciting way to mediate their intrinsic ferromagnetism (FM) with strain engineering. Here, we summarize the recent progress of strain engineering of intrinsic FM in 2D van der Waals materials. First, we in-troduce how to explain the strain-mediated intrinsic FM on Cr-based and Fe-based 2D van der Waals materials through ab initio Density functional theory (DFT), and how to calculate magnetic anisotropy energy (MAE) and Curie temperature (TC) from the interlayer exchange coupling J. Subsequently, we focus on numerous attempts to apply strain to 2D materials in experiments, including wrinkle-induced strain, flexible substrate bending or stretching, lattice mismatch, elec-trostatic force and field-cooling. Last, we emphasize that this field is still in early stages, and there are many challenges that need to be overcome. More importantly, strengthening the guideline of strain-mediated FM in 2D van der Waals materials will promote the development of spintronics and straintronics.”

[Comment]:

Point 2: In the actual stage, I do not recommend the manuscript for publication. If the authors want to improve the manuscript. I strongly recommend to investigate further the literature about CrI3. There is a lot of publications (below I suggest some) that can help the authors to have a more complete perspective about the actual understanding of strain in 2D ferromagnets, to build the first part of the paper.

Banasree Sadhukhan, Anders Bergman, Yaroslav O Kvashnin, Johan Hellsvik, and Anna Delin. Spin-lattice couplings in two-dimensional CrI3 from first-principles computations. Physical Review B, 105(10):104418, 2022.

Esteras, D. L., Rybakov, A., Ruiz, A. M., & Baldoví, J. J. (2022). Magnon straintronics in the 2D van der Waals ferromagnet CrSBr from first-principles. Nano Letters, 22(21), 8771-8778.

Kashin, I. V., Mazurenko, V. V., Katsnelson, M. I., & Rudenko, A. N. (2020). Orbitally-resolved ferromagnetism of monolayer CrI3. 2D Materials, 7(2), 025036.

Soriano, D., Rudenko, A. N., Katsnelson, M. I., & Rösner, M. (2021). Environmental screening and ligand-field effects to magnetism in CrI3 monolayer. npj Computational Materials, 7(1), 162.

Kvashnin, Y. O., Bergman, A., Lichtenstein, A. I., & Katsnelson, M. I. (2020). Relativistic exchange interactions in CrX3 (X= Cl, Br, I) monolayers. Physical Review B, 102(11), 115162.

[Response]:

We thank you very much for your insightful comments and providing this important literature. According to your suggestion, we have rewritten the first part of the paper (Introduction) and added the five references as Refs. 92, 95, 93, 94 and 96 at the beginning of section "Theoretical calculations" in our revised manuscript.

“According to the Mermin-Wanger-Hohenberg theory [1, 2], thermal fluctuations can destroy the long-range magnetic order of 2D systems at finite temperature. However, the anisotropy of the system suppresses thermal disturbances by opening the gap in the spin-wave spectrum [3-6]. Furthermore, spin orbit coupling (SOC) can stabilize the long-range magnetic order in 2D systems by contributing to magnetic anisotropy. After the discovery of low-temperature long-range ferromagnetic order in monolayer Cr2Ge2Te6 and CrI3 [7, 8], many efforts have been made to achieve a room temperature (RT) ferromagnet. Indeed, strain engineering [9-25] is a very important strategy for mediating material properties, including optoelectronic [9, 10, 13-15, 21], electrocata-lytic [11, 16, 22-24], and magnetic properties [15, 19, 21, 24-28]. Since Novoselov et al. [29] obtained stable monolayer graphene in the laboratory in 2004, further research gradually revealed that 2D materials such as MoS2 could withstand up to 20% strain [30-33]. However, it was very difficult to directly apply strain to 2D materials in experiments, which made the strain-controlled performance largely re-main in theoretical study. This was because by changing lattice parameters, strain could be easily applied to the lattice of 2D materials. Specifically, the study on strain-mediated magnetism in 2D materials, especially in 2D van der Waals materials with intrinsic long-range FM order, was focused on theoretical calculation. More interestingly, the Bi2Te3|Fe3GeTe2 heterostructure related to strain [34, 35], have been designed to increase the Curie temperature (TC) due to the proximity effect [36-40].

Very recently, some significant progress had also been made in the field of experimental research [27, 28, 41-48]. Here, we summarize the recent progress of strain engineering of intrinsic ferromagnetism (FM) in 2D van der Waals materials in Figure 1. First, we introduce how to explain the strain-mediated intrinsic FM on Cr-based and Fe-based 2D van der Waals materials with long-range FM order through ab initio Density functional theory (DFT), and how to calculate magnetic anisotropy energy (MAE) and Curie temperature from the interlayer exchange coupling J. Subsequently, we focus on numerous attempts to apply strain to 2D materials in experiments, including wrinkle-induced strain, flexible substrate bending or stretching, lattice mismatch, elec-trostatic force and field-cooling. Last, we emphasize that this field is still in early stages, and there are many challenges that need to be overcome. More importantly, strengthening the guideline of strain-mediated FM in 2D van der Waals materials will promote the development of spintronics [6, 49-57] and straintronics [12, 19].”

“In order to understand the essence of 2D ferromagnetism, ab initio Density functional theory, including linear density approach (LDA) [59], local spin density ap-proximation (LSDA) [92], the generalized gradient approach (GGA) [61, 93, 94], and DFT+U [95, 96] was often used to calculate the electronic structure of the system as a starting point. Moreover, the interlayer exchange coupling J was closely related to magnetic anisotropy, and it would also be used to calculate TC [86, 97-100]. The mean field theory [97, 101] would roughly estimate TC, but TC often was overestimated it. Although the random phase approximation (RPA) could more accurately estimate TC of three-dimensional (3D) materials, it may fail in 2D systems with large anisotropy. Notably, classic Monte Carlo (MC) [6, 97] simulations can also well describe the critical temperature.”

[Comment]:

Point 3: The redaction is not attractive and focuses in small and technical details like the Curie temperature being increased 5K with strain without any rationalization that shows the understanding of ferromagnetism and strain that the community has nowadays. About the section 3, I support the structure and the idea of explaining the main techniques to perform strain 2D materials, however the writing should be focused in rationalizing and explaining the techniques and their potential and limitations, supported by the experiments in bibliography. At the moment this section just rewrites the explanation of experiments that were already explained in their respective papers, and does not provide a significative rationalization. Neither extracts a perspective or a big picture that explains how all these experiments contributed to the field.

At least the contents of the 3.1 version were already explained in their previous Nanomaterials (citation 18) and the authors present a rewritten version.

About the section 3, I strongly recommend to read some reviews and papers related to the latest experiments in strain of 2D materials, this can help to extract a big picture and improve the storytelling of this second part of the paper.

[Response]:

Thank you very much for your comment. According to your suggestion, we have deleted and revised the relevant description at the section 3.1 on page 9-10.

“The previously discovered strain-mediated FM phenomena were all in 2D materials without intrinsic long-range magnetic order [27, 28, 41]. Very recently, seidel et al. [36] found a clear dependence of TC on the strain state and the thickness of Cr2Ge2T6 (CGT). The layered CGT lattice with intrinsic magnetism was shown in Figure 9(a). A stripe domain structure was observed with MFM (Figure 9b). After buckling, the wrinkled area had a higher TC than the flat area on (Figure 9c). DFT calculations indicated that the strain could elevate the TC in monolayer and bilayer Cr2Ge2T6 (Figure 9d-g). In addition, the strain of three different wrinkles in Figure 9(h) was confirmed by COMSOL simulations. To the best of our knowledge, this was the first time that the strain distribution in CGT wrinkles has been mapped with COMSOL simulation. Furthermore, the magnetic signals could be observed at the wrinkled regions at RT through MFM.”

In response to your comment, we have rewritten the first and last part of the paper (Introduction and Conclusions) and improved the storytelling of this second part of the paper (Theoretical calculations). Specifically, we will introduce the progress in theoretical calculations from the perspectives of Cr-based and Fe-based 2D van der Waals materials on page 2-9.

“In this review, we have summarized the recent progress of strain-mediated intrinsic FM in 2D van der Waals materials with long-range order. First, we introduce how to explain the strain-mediated intrinsic FM on Cr-based and Fe-based 2D van der Waals materials with long-range FM order through ab initio Density functional theory (DFT), and how to calculate magnetic anisotropy energy (MAE) and TC from the interlayer exchange coupling. Subsequently, we focus on numerous attempts to apply strain to 2D materials in experiments, including wrinkle-induced strain, flexible substrate bending or stretching, lattice mismatch, electrostatic force and field-cooling. However, research in this field is still in its early stages and there are many challenges that need to be overcome.

By selecting substrates with different thermal expansion coefficients, including x-quartz [135-138], sapphire [27, 28, 139], and SiO2 [114]), uniaxial [135-138] or biaxial strain [27, 28, 114, 139, 140] could be introduced into 2D materials. The introduction of biaxial strain in this way was limited to the study of 2D materials without intrinsic long-range magnetic order such as MoS2 [28] and ReS2 [27]. So far, this strategy has not been applied to the research on 2D intrinsic van der Waals materials. Very few equipment [47, 48, 121] could apply strain to 2D materials, but less could be coupled with magnetic testing equipment [41, 141]. Especially, it is very challenging that one could accurately determine the strain while measuring the magnetic properties and TC in different regions.

Recently, some emerging magnetic imaging technologies such as magneto-optical Kerr effect (MOKE) [7, 8, 40, 142-144] , magnetic circular dichroism (MCD) [62, 126, 145-149], photoemission electron microscopy (PPEM) [150], scanning trans-mission X-ray microscopy (STXM) [151], Lorentz transmission electron microscopy (LTEM) [150-152], spin-polarized scanning tunneling microscopy (SP-TEM) [66, 153], MFM [41, 154], scanning SQUID [155-159], and scanning nitrogen-vacancy center microscopy (SNVM) [160-164] have been used to study 2D FM materials. However, these studies have not yet been combined with strain, and currently it was almost impossible to directly analyze the relationship between strain and FM in 2D intrinsic materials. More importantly, strengthening the guideline of strain-mediated FM will promote the development of spintronics and straintronics.”

“2.1. Cr-based 2D van der Waals materials

2.1.1. CrCl3; 2.1.2. CrBr3; 2.1.3. CrI3; 2.1.4. CrTe2; 2.1.5. Cr2Ge2Te6; 2.1.6. CrPS4

2.2. Fe-based 2D van der Waals materials”

[Comment]:

Point 4: The design of the figures is also very poor; the Figures are enormous panels that don’t even enter in the margins of the manuscript. They can show between 4-7 times the same structure of the material, given the message of the figures was not rationalized properly.

[Response]:

Thank you very much for your comment. In response to your comment, we have adjusted the size and order of the figures to make them more reasonable. In addition, we have added the relevant description.

“To the best of our knowledge, this was the first time that the strain distribution in CGT wrinkles has been mapped with COMSOL simulation.” on Line 321-323.
